# A Knowledge-Injected Curriculum Pretraining Framework for Question Answering

## ABSTRACT

Knowledge-based question answering (KBQA) is a key task in natural language processing research, and also an approach to access the web data and knowledge, which requires exploiting knowledge graphs (KGs) for reasoning. In the literature, one promising solution for KBQA is to incorporate the pretrained language model (LM) with KGs by generating KG-centered pretraining corpus, which has shown its superiority. However, these methods often depend on specific techniques and resources to work, which may not always be available and restrict its application. Moreover, existing methods focus more on improving language understanding with KGs, while neglect the more important human-like complex reasoning. To this end, in this paper, we propose a general **K**nowledge-**I**njected **C**urriculum **P**retraining framework (KICP) to achieve comprehensive KG learning and exploitation for KBQA tasks, which is composed of knowledge injection (KI), knowledge adaptation (KA) and curriculum reasoning (CR). Specifically, the KI module first injects knowledge into the LM by generating KG-centered pretraining corpus, and generalizes the process into three key steps that could work with different implementations for flexible application. Next, the KA module learns knowledge from the generated corpus with LM equipped with an adapter as well as keeps its original natural language understanding ability to reduce the negative impacts of the difference between the generated and natural corpus. Last, to enable the LM with complex reasoning, the CR module follows human reasoning patterns to construct three corpora with increasing difficulties of reasoning, and further trains the LM from easy to hard in a curriculum manner to promote model learning. We provide an implementation of the general framework, and evaluate the proposed KICP on four real-word datasets. The results demonstrate that our framework can achieve higher performances, and have good generalization ability to other QA tasks.

## KEYWORDS

Question answering, Knowledge-injected pretraining, Curriculum learning

## 1 INTRODUCTION

Knowledge-based question answering (KBQA) is a key task in natural language processing and data mining research [27], which could act as an approach to access and process web data and knowledge,

*Conference acronym 'XX, June 03–05, 2018, Woodstock, NY*
© 2018 Association for Computing Machinery.
ACM ISBN 978-1-4503-XXXX-X/18/06...$15.00
https://doi.org/XXXXXXX.XXXXXXX

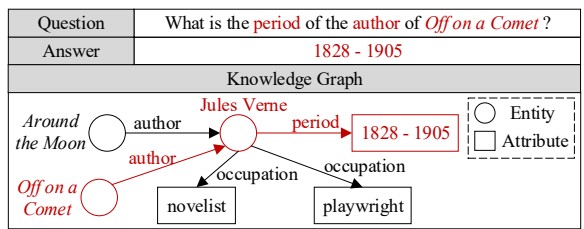

| Question | What is the period of the author of *Off on a Comet* ? |
|---|---|
| Answer | 1828 - 1905 |

**Figure 1: A toy example of KBQA, which requires complex reasoning marked in red.**

and lead to useful applications such as smart voice assistant and search engine especially with the large language models [24]. As shown in Figure 1, KBQA aims to answer questions in natural language based on background knowledge, which is often formatted as knowledge graphs (KGs) [39, 43]. Therefore, KBQA requires abilities of both natural language understanding (NLU) and knowledge reasoning, making it a challenging task in related fields.

In the literature, researchers have proposed many solutions for KBQA [21, 27, 43], among which the pretrained language models (LMs) have become the most promising for its strong NLU ability [6, 24]. Unfortunately, LMs work not so well in knowledge application [15, 17], which hinders its application in KBQA. Therefore, researchers have tried great efforts to enhance the LMs with KGs (inputting knowledge facts into LMs, or pretraining LMs with knowledge-based tasks [15, 25, 28, 31, 32, 40, 41, 44, 46]), which has greatly improved LMs in knowledge-related tasks. However, these methods often learn KGs as supplementary to additional pretraining corpus [15, 44], which can not cover the whole KG and may overlook some knowledge useful in certain tasks, and thus leads to incomplete knowledge learning. Towards this point, a straightforward solution is to generate the pretraining corpus based on the KGs. Although many methods have been developed along this line [1, 3, 14, 42], they usually depend on specific techniques or resources for effective corpus generation (e.g., requiring pretrained generative model to generate sentences, or generating sentences in a fixed format), which may be unavailable in practice and thus restricts its application. Therefore, in this paper we hope to design a general framework to generate KG-centered corpus for comprehensive knowledge pretraining of LMs, which could work with different detailed techniques for flexible application.

However, along this line there exist several nontrivial technical challenges. First, there are many solutions to generate sentences based on given KGs for different demands (e.g., pretrained generative LMs [1], fixed sentence templates [14]). Moreover, although most KGs store the knowledge triplets with entity IDs, some high-quality KGs also contain additional attribute information, which is stored in various forms (e.g., texts, numbers and dates) and requires different processing. How to unify and generalize these various techniques and data forms remains much open. Second, the

generated sentences differ from natural ones and may even seem distorted, which may mislead the LM and hurt natural language understanding ability of the LM in pretraining. Existing methods address this problem with specific techniques in accordance with their generation methods (e.g., generating sentences more similar to natural ones with complex generative LMs [1], or adopting specially designed sentence templates to reduce the negative impacts [14]), but how to overcome this shortcoming for an arbitrary generation method in the general framework is a nontrivial problem. Last, existing methods enhancing LMs with KGs [32, 44] focus more on improving language understanding with related knowledge, while seldom have considered the human-like complex reasoning ability. Humans can perform reasoning over multiple knowledge facts following specific patterns, which is also widely required in KBQA tasks. For example, in Figure 1, to reach the answer, the LM first needs to find that the author of *Off on a Comet* is Jules Verne, and then the period of Jules Verne is 1828-1905. How to enable the LMs with such complex reasoning is a challenging problem.

To this end, in this paper, we propose a general **K**nowledge-**I**njected **C**urriculum **P**retraining framework (KICP) to achieve comprehensive KG learning and exploitation for KBQA, which is composed of knowledge injection (KI), knowledge adaptation (KA) and curriculum reasoning (CR). Specifically, the KI module converts KG triplets into sentences to construct pretraining corpus for complete knowledge learning, and generalizes the process into three key steps, i.e., text characterization, sentence construction and masking, which can be implemented with different detailed techniques and various data forms for flexible application. Next, to reduce the negative impacts brought by the difference between generated and natural corpus on LM pretraining, the KA module fixes the original LM to keep its NLU ability, and learns knowledge from the generated corpus with a trainable adapter working with the LM. Last, to pretrain the LM with complex reasoning ability, the CR module follows common reasoning patterns of humans and constructs corpora requiring complex knowledge reasoning. Furthermore, the CR module arranges the complex corpora into three lessons with increasing difficulties, and trains the LM from easy to hard following the curriculum learning manner to reduce pretraining difficulty. Finally, we provide an implementation of the general framework, and conduct extensive experiments on four real-word datasets to evaluate KICP. The results demonstrate that our framework can achieve higher performances, and generalize to other QA tasks well.

## 2 RELATED WORK

**Knowledge-Based Question Answering.** Knowledge-based question answering (KBQA) aims to answer questions based on given knowledge bases, which are usually knowledge graphs (KGs) [13, 21, 27, 43]. In the literature, studies on KBQA can be roughly divided into two branches, i.e., the knowledge-enhanced LM centered on question understanding and reasoning with LMs (which is the focus of this paper, and we will provide a more detailed introduction later), and KG-based reasoning centered on performing knowledge reasoning on the graph, which includes path-based [20], embedding-based [9, 27] and graph-based methods [8, 12, 21, 38, 39, 43]. Path-based methods map the question into entities and relations and

perform reasoning by directly walking on the KG to reach the answers [20], which have higher interpretability but require much effort on rule design in complex questions. Embedding-based methods such as EmbedKGQA [27] represent the question and KG in the same latent space, and infer the answer with simple vector computation. These methods unify simple and complex reasoning, but may have limited performance and interpretability. Graph-based methods are widely studied recently [8, 12, 21, 38, 39, 43], which sample a sub-graph from the KG, and perform detailed reasoning on the sub-graph with neural networks. Graph-based methods are widely applied in complex reasoning for the good trade-off between interpretability, performance and computation complexity, but the knowledge modeling is insufficient only within the sub-graph which may leads to limited robustness.

**Knowledge-Enhanced Language Model.** As the pretrained language models demonstrate great performances in natural language processing [4, 23, 24, 30], their shortcomings on knowledge-based tasks are also exposed [15, 17]. Therefore, researchers have tried many efforts to enhance LMs with knowledge from KGs, which could be roughly divided into explicit methods [15, 25, 44] and implicit methods [14, 28, 31, 32, 37]. Explicit methods feed knowledge facts or embeddings into LM as additional inputs to exploit related knowledge. For example, K-BERT [15] injected the knowledge triplets into the sentences as inputs to the LM. Zhang *et al.* [44] developed an aggregator network to incorporate the semantic vectors learned by LM and entity embeddings from KGs. Implicit methods design special pretraining tasks to learn knowledge from KGs and corpus with LM. To better learn entity knowledge from corpus, Sun *et al.* [28] introduced an entity masking strategy for masked language model pretraining, and Wang *et al.* [32] trained LM as knowledge embedding model by encoding descriptions of entities with LM as embeddings. To better exploit the multilingual triplets of the KG, Liu *et al.* [14] generated multilingual synthetic corpus using the KG triplets and pretrained the LMs, and Agarwal *et al.* [1] designed a more complicated pipeline with pretrained generative LM to synthesize more natural corpus based on KGs. In summary, explicit methods can exploit the knowledge in a more direct manner but require more knowledge annotations as additional inputs, while implicit methods can be easily applied in downstream tasks, but require heavy pretraining for each knowledge base.

**Curriculum Learning.** Curriculum learning is an effective continual optimization strategy first proposed by Bengio [2], which imitates human learning habits starting by easy lessons and then more difficult ones, and demonstrates that training model on datasets from easy to hard could benefit learning, accelerate convergence and promote the training outcome. Curriculum learning has shown great superiority in improving the generalization and convergence of models, and has been widely applied in various fields [11, 22, 45]. For example, Zhao *et al.* [45] designed pretraining tasks with different difficulties and applied curriculum learning to train a LM for mathematics understanding, and Li *et al.* [11] trained the visual question solver on a sequence of instance sets with increasing complexity following the curriculum manner.

Our work differs from previous methods as follows. First, existing methods rewriting the KG into corpus often depend on specific techniques and resources for effective generation, while our method is a general framework which can work with different

detailed implementations for flexible application under different circumstances. Second, existing methods focus more on improving language understanding with related knowledge but seldom consider the human-like complex reasoning ability of LMs, while our method explicitly enables the LM with such ability with specially designed pretraining task and further adopt the curriculum learning strategy to promote the outcome.

## 3 KICP: KNOWLEDGE-INJECTED CURRICULUM PRETRAINING

In this section, we first formally introduce the KBQA task, and then present the proposed KICP framework.

### 3.1 Problem Definition

Knowledge-based question answering (KBQA) is composed of the knowledge graph $\mathcal{KG}$ and the question-answer pair $(Q, Y)$. Without loss of generality, we suppose that the KG contains knowledge triplets about the relation between two entities and the attribute of each entity where the attribute values are in diverse forms but can be expressed with texts anyway (natural language texts are defined as $V^+$, where $V$ is the vocabulary). Therefore, the KG can be defined as $\mathcal{KG} = (\mathbb{E}, \mathbb{R}, \sum)$, where $\mathbb{E}$ is the entity set, $\mathbb{R}$ is the relation and attribute set, and $\sum$ means the knowledge triplets. Each triplet $(h, r, t) \in \sum (h, t \in \mathbb{E}, r \in \mathbb{R})$ means that the entity $h$ and $t$ have the relation $r$ (e.g., "Jules Verne" is the "author" of "*Off on a Comet*" in Figure 1), and $(h, r, t) \in \sum (h \in \mathbb{E}, r \in \mathbb{R}, t \in V^+)$ means the attribute $r$ of entity $h$ is $t$, where $t$ is the attribute value in text (e.g., the "period" of "Jules Verne" is "1828-1905") Besides, each entity $e \in \mathbb{E}$ is assigned with several names in natural language $N_e = \{n_{e1}, n_{e2}, \ldots, n_{ek}\}$ (each name $n_{ei} \in V^+$). $\mathbb{R}$ is assigned with names similarly. In the question-answer pair $(Q, Y)$, $Q = \{q_1, q_2, \ldots, q_n\} \in V^+ (q_i \in V)$ is the question in natural language, and $Y$ is the answer to $Q$ inferred under $\mathcal{KG}$, whose form depends on the task (e.g., most KBQA selects an entity or attribute value from $\mathcal{KG}$ like Figure 1, and generative QA generates formal language from certain vocabulary such as natural text or mathematical expression).

Given the knowledge graph $\mathcal{KG}$ and question-answer pair $(Q, Y)$, the goal of KBQA is to train a model $M : (\mathcal{KG}, Q) \rightarrow Y$ to predict the answer $Y$ of question $Q$ under $\mathcal{KG}$. In this paper, we first pretrain a language model $\mathcal{LM}$ with $\mathcal{KG}$, and then use it in $M$ to predict the answer $Y$ to $Q$. We expect that $\mathcal{LM}$ could learn knowledge from $\mathcal{KG}$ comprehensively and well handle complex reasoning.

### 3.2 Method

We propose a general **K**nowledge-**I**njected **C**urriculum **P**retraining framework (KICP) to pretrain $\mathcal{LM}$ for comprehensive knowledge learning and complex reasoning, which could easily work with different detailed implementations for flexible applications. As shown in Figure 2 (a), KICP is composed of three key components, i.e., *knowledge injection* (KI), *knowledge adaptation* (KA) and *curriculum reasoning* (CR). Specifically, KI injects knowledge from the KG into the LM completely by converting the KG triplets to sentences to construct the pretraining corpus, and generalize the various generation techniques into three key steps. To reduce the negative impacts brought by the gap between generated and natural corpus,

KA fixes the original LM to keep its NLU ability, and equips the framework with a trainable knowledge adapter to learn knowledge from the generated corpus. To pretrain the LM with complex reasoning ability, CR follows common patterns of human reasoning and constructs several reasoning-required corpora with different difficulties, and trains the LM from easy to hard in a curriculum manner to promote model learning.

*3.2.1 Knowledge Injection.* To overcome the insufficient knowledge learning brought by using the KG as supplementary to external corpus, we directly convert the KG triplets into sentences as pretraining corpus to inject knowledge into the LM. Moreover, there exist several effective sentence generation techniques for different requirements in the literature [1, 14], and the KGs contain multiple forms of data that requires different processing (e.g., IDs, texts, numbers and dates). Therefore, to generalize these detailed techniques to a general framework for flexible application in various circumstances, as shown in Figure 2 (b), we abstract the sentence generation process into three key steps, i.e., text characterization, sentence construction and masking.

**Text Characterization.** Given one triplet $k = (h, r, t) \in \sum$ sampled from $\mathcal{KG}$, KI first characterizes all fields of the triplet as texts (*Txt*), which serve as the backbone elements of the sentence to generate. For the entities and relations stored in IDs, We map the meaningless ID (e.g., e1) to a meaningful name (Jules Verne), which is dynamically sampled from the associated name set in each iteration to increase corpus diversity. More sampling strategies can also be applied here for other demands [14]. For the various forms of attribute values (e.g, numbers, dates and texts), we use their textual descriptions as they can always be expressed with texts despite the original forms. In this way, we can unify the diverse processing of the entities, relations and attribute values.

**Sentence Construction.** After getting the textual elements, KI applies a sentence construction strategy $\tau$ to assemble these elements into a complete sentence, including reordering and transforming the elements and adding auxiliary words. The strategy $\tau$ can be implemented with different existing techniques, such as sentence templates, grammar-based rules, and the generative LMs [1, 14].

**Masking.** The last step is to mask the generated sentence for masked language model (MLM) pretraining. To force knowledge learning and match the differences between entities and attribute values, we prefer paying more weights to the knowledge elements in the sentence (those converted from the triplet), and applying different masking strategies *Msk* to entities and attribute values. For example, we apply the entity masking [28] on entities which masks the whole entity name to force learning relation knowledge instead of memorizing the entity name, and whole word masking (WWM) [5] on attribute values since the values may contain too much information (e.g., biography) and are too hard to recover if all masked. WWM also works similarly to entity masking on short values (e.g., numbers) by masking as a whole word. More masking techniques can be used here as *Msk*.

Overall, the sentence generation process is formulated as follows:

$$KI(k) = Msk(\tau(Txt(h), Txt(r), Txt(t))), \quad k = (h, r, t) \in \sum. \quad (1)$$

The knowledge-injected corpus is composed of the sentences $KI(k)$, which are dynamically generated from triplets sampled from the

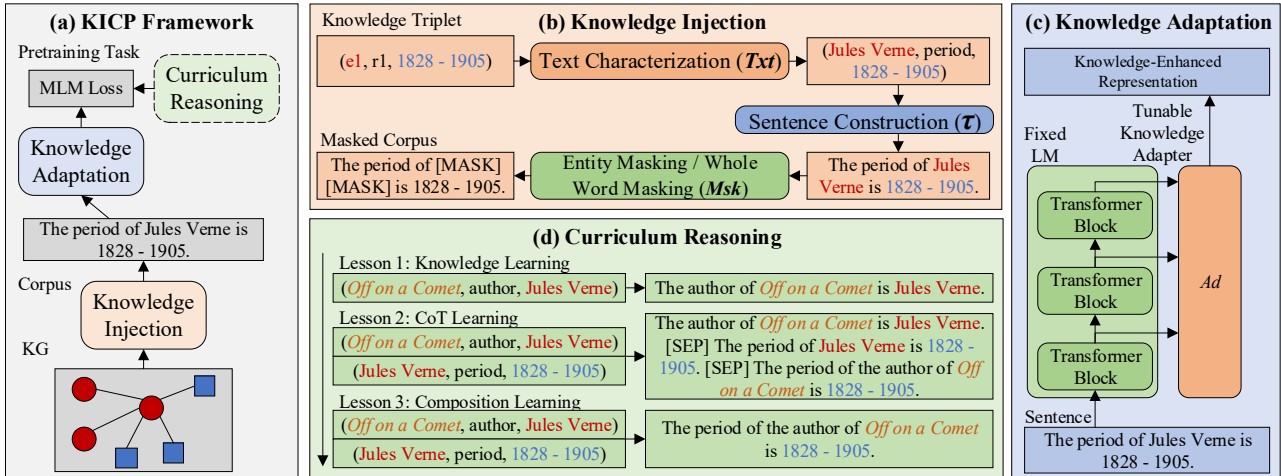

**Figure 2: The architecture of the proposed KICP framework. (a) The overview of KICP. (b) The knowledge injection module (KI) converts the KG triplets into sentences with three key steps. (c) The knowledge adaptation module (KA) works with the LM to keep original NLU ability as well as learn knowledge. (d) The curriculum reasoning module (CR) constructs complex reasoning-required corpora to pretrain the LM from easy to hard.**

KG in pretraining. Compared with existing methods rewriting KG as corpus, KI does not depend on specific techniques or resources, and thus could work with different implementations for various application demands.

*3.2.2 Knowledge Adaptation.* Obviously the corpus generated by KI differs from natural ones as the sentences may not strictly follow the grammar (especially for some simple $\tau$), and the diversity of the corpus is limited. Pretraining the LM on the corpus may hurt NLU ability and work badly on natural texts. Furthermore, as the sentence generation technique in the proposed general framework is arbitrary, we can not use methods associated with specific generation techniques to address the problem as existing studies [1, 14]. Therefore, in knowledge adaptation (KA), we turn to keeping the NLU ability of LM during knowledge pretraining.

As demonstrated by Figure 2 (c), following the adapter paradigm in LM tuning [7, 31], we fix the LM parameters and add a trainable knowledge adapter module $Ad$ above the original LM $LM$. $Ad$ uses the semantic outputs of $LM$ as inputs, and outputs the knowledge-enhanced representations. Moreover, to deeply improve the fusion of the semantics and knowledge, the semantic outputs of all layers in the LM are used. The computation of KA is formulated as follows:

$$KA(x) = Ad(LM(x)), \quad (2)$$

where $x$ is the input sentence. $Ad$ can be implemented with any neural networks, which is expected to have a proper size to contain enough space for knowledge learning and avoid greatly increasing computation complexity as well.

In pretraining, the parameters of $Ad$ is trained to learn knowledge from the constructed corpus, while the original LM is fixed. As the original LM is not affected by $Ad$, the NLU ability is retained as much as possible to reduce the negative impacts of the gap between generated and natural corpus.

*3.2.3 Curriculum Reasoning.* With KI and KA, KICP can effectively inject the KG into LM, but still lacks complex reasoning ability over multiple knowledge facts as required in real-world KBQA tasks. To enable the LM with such ability, the curriculum reasoning module (CR) pretrains LM on corpora requiring complex reasoning as shown in Figure 2 (d).

It is hard to collect enough reasoning-required corpus for all KGs, so we also build the corpus based on the KG. Humans often perform complex reasoning following specific patterns (e.g., multi-top reasoning), which put restrictions on the participating triplets (e.g., the chain-like triplets). Therefore, we build the corpus following these patterns (e.g., "The period of the author of *Off on a Comet* is 1828-1905"). We first sample several triplets $\{k_1, \ldots, k_n\}$ matching the restrictions from KG, such as the chain-like triplets $\{(Off on a Comet,$ author, Jules Verne$), (Jules Verne,$ period, 1828-1905$)\}$ for multi-hop reasoning, and then convert them into a complex composition with a pipeline $Comp$ similar to KI as follows:

$$Comp(k_1, \ldots, k_n) = Msk'(\tau'(Txt(h_1), Txt(r_1), Txt(t_1),$$
$$\ldots, Txt(t_n))), \quad k_i = (h_i, r_i, t_i) \in \sum, \quad (3)$$

where $\tau'$ and $Msk'$ are sentence construction and masking in $Comp$. Much more reasoning patterns can be supported by the CR module.

The complex composition often discards some information to infer from knowledge, so it is hard to pretrain LM directly (e.g., in previous example "Jules Verne" is discarded, which makes it hard to understand without related knowledge). Therefore, as shown in Figure 2 (d), we split the pretraining into three lessons with generated corpora from easy to hard following curriculum learning [45] to promote model learning.

**Lesson 1: Knowledge Learning.** We start by pretraining LM on single triplets from the KG. We build this corpus with KI based on one triplet $k$ for each sentence, and pretrain the LM (i.e., KA) on the MLM task to memorize the knowledge facts as follows:

$$\min_{\theta_{Ad}, \theta_{MLM}} L_1(k) = MLM(KA(KI(k))), \quad (4)$$

where $\theta_{Ad}$ and $\theta_{MLM}$ means trainable parameters for knowledge adapter $Ad$ in $KA$ and MLM head.

**Lesson 2: CoT Learning.** Having learned basic knowledge facts from KG, next we teach the LM how to conduct complex reasoning with related knowledge facts. Inspired by chain-of-thought (CoT) [19, 34], we assemble each sentence with complex composition by $Comp$ for certain reasoning pattern and all related knowledge by $KI$ as reasoning steps base on triplets $\{k_1, \ldots, k_n\}$. To avoid information leakage, we mask the same element (e.g., entity) in both the final composition and reasoning steps, and pretrain the LM on the MLM task as follows:

$$\min_{\theta_{Ad},\ \theta_{MLM}} L_2(k_1, \ldots, k_n) = MLM(KA([KI(k_1), \ldots,$$
$$KI(k_n), Comp(k_1, \ldots, k_n)])), \quad (5)$$

where $[,]$ means text concatenation, and $\{k_1, \ldots, k_n\}$ matches the reasoning pattern for $Comp$.

**Lesson 3: Composition Learning.** In the hardest lesson, we pretrain the LM to reason with memorized knowledge as real-world QA tasks, where we only provide the final compositions without related reasoning steps. Therefore, We construct the corpus with the complex compositions by $Comp$, and pretrain the LM on the MLM task as follows:

$$\min_{\theta_{Ad},\ \theta_{MLM}} L_3(k_1, \ldots, k_n) = MLM(KA(Comp(k_1, \ldots, k_n))). \quad (6)$$

The corpora are dynamically generated with randomly sampled triplets in pretraining. We demonstrate some samples of corpora in three lessons in Appendix C. Through the three pretraining lessons, we explicitly enable the LM with human-like complex reasoning ability required in KBQA tasks, and reduce the pretraining difficulty with the curriculum learning.

*3.2.4 QA Fine-Tuning.* After pretrained on the KG, the LM can be easily applied in different downstream QA tasks without additional annotations or external knowledge inputs. Specifically, the LM (i.e., $KA$) reads the question $Q$ as input, and outputs the knowledge-enhanced vector, which is fed to a task-dependent prediction head $Pred$ to generate the answer $Y$. The whole system ($LM$ and $Ad$ in $KA$ and $Pred$) can be fine-tuned on different QA tasks subject to the task-dependent objective function $\mathcal{L}$ as follows:

$$\min_{\theta_{LM},\ \theta_{Ad},\ \theta_{Pred}} L_{QA}(Q, Y) = \mathcal{L}(Pred(KA(Q)), Y), \quad (7)$$

where $\theta_{LM}$, $\theta_{Ad}$ and $\theta_{Pred}$ are parameters of these modules.

## 3.3 Implementation

In this section, we provide an implementation of the general KICP framework. In KI, we implement text characterization and masking as mentioned in section 3.2.1, and realize $\tau$ by simply concatenating all fields, which works well on our datasets.

In KA, we implement the knowledge adapter $Ad$ as BERT with the same number of layers and halved vector dimension. In each layer of $Ad$, the input (semantic vector from corresponding layer of $LM$) is first projected with a linear model to the latent space of hidden vector from last layer, and then added with the hidden vector to feed to the BERT layer. The final vectors of $Ad$ and $LM$ are merged with a linear layer as the output. The architecture of KA is available in Appendix A.

In CR, we implement $Comp$ with two widely-used reasoning patterns, i.e., multi-hop reasoning and multi-object reasoning. **Multi-hop reasoning** (e.g., the period of the author of *Off on a Comet* is 1828-1905) first infers an intermediate entity from the topic entity in the question (the author of *Off on a Comet* is Jules Verne), and then use it to infer the next intermediate entity until reaching the answer (the period of Jules Verne is 1828-1905). Therefore, the knowledge triplets form a chain-like structure, where the tail entity of one triplet is the head of the next one (e.g., Jules Verne). Given these triplets, $Comp$ discards all intermediate entities and concatenates other fields sequentially. **Multi-object reasoning** (e.g., the occupation of Jules Verne is novelist and playwright) infers several results from one topic entity, thus the knowledge triplets share the same head entity and relation (Jules Verne and occupation). Given the triplets, $Comp$ discards the heads and relations expect the first one, and concatenates all tails with the first head and relation. For each sentence we sample 2 to 3 triplets matching the patterns.

## 4 EXPERIMENTS

In this section, we conduct experiments on four QA datasets to evaluate the proposed KICP framework. *We will release our code and public data after this paper is accepted.*

### 4.1 Experimental Setup

*4.1.1 Datasets.* We use three KBQA datasets, i.e., CN-QA, ComplexWebQuestions and FreebaseQA, to evaluate KICP on knowledge-based reasoning, and a generative dataset Math23K for generalization to other knowledge-related QA tasks.

**CN-QA** is a Chinese KBQA dataset collected from smart voice assistant accompanied by a KG named CN-KG with both entity relations and attributes. **ComplexWebQuestions** [29] is a public KBQA dataset with complex questions built on WebQuestions and Freebase. **FreebaseQA** [10] is another public KBQA dataset based on Freebase with both simple and complex questions derived from TriviaQA and trivia websites. Since Freebase has been merged to Wikidata, we use the Wikidata dump in [32], and map entities to Wikidata to construct an answerable subset for ComplexWebQuestions and FreebaseQA. **Math23K** [33] is a public generative math word problem dataset which answers the question with a generated mathematical expression. We construct a KG based on the semantic web HowNet [26] for Math23K following [35].

The questions in KBQA are answered with entities or attribute values from the KG. To reduce the computation complexity without losing much difficulty, we sample 10 hard candidate answers with the same type of the truth for three KBQA datasets, among which the prediction is made. We also sample a sub-graph from the whole KG for each dataset to accelerate pretraining. The statistics of the datasets are available in Appendix B.

*4.1.2 Baseline Methods.* We compare the proposed KICP with original LMs **BERT** [6] and **RoBERTa** [16], and knowledge-enhanced LMs **ERNIE** [44], **K-BERT** [15], **KEPLER** [32] and **K-Adapter** [31]. A brief introduction to these baselines are listed as follows.

- **BERT** [6] was the most widely used pretrained language model, based on which our framework is implemented, thus we add BERT as baseline to evaluate the improvement.

**Table 1: Overall Results of All Methods on Four Datasets**

| Dataset | CN-QA | | ComplexWebQuestions | | FreebaseQA | Math23K |
|---|---|---|---|---|---|---|
| Metric | F1 | EM | F1 | EM | ACC | ACC |
| BERT | 0.607 | 0.458 | 0.856 | 0.763 | 0.896 | 0.801 |
| RoBERTa | 0.610 | 0.456 | 0.863 | 0.779 | 0.892 | 0.803 |
| ERNIE | 0.614 | 0.459 | 0.861 | 0.772 | 0.901 | 0.796 |
| K-BERT | 0.620 | 0.462 | 0.866 | 0.774 | 0.896 | 0.799 |
| KEPLER | 0.628 | 0.467 | 0.868 | 0.785 | 0.906 | / |
| K-Adapter | 0.612 | 0.462 | 0.866 | 0.802 | 0.905 | / |
| KICP-KA | 0.633 | 0.469 | 0.871 | 0.809 | 0.903 | 0.797 |
| KICP-ATT | 0.629 | 0.466 | / | / | / | / |
| KICP | **0.639*** | **0.480*** | **0.880*** | **0.819*** | **0.911*** | **0.809*** |

- **RoBERTa** [16] studied the impacts of hyperparameters and task design in pretraining, and achieved a robustly optimized BERT with significant improvements.
- **ERNIE** [44] developed an aggregator network to explicitly combine the entity embedding learned from KG with the semantics learned by LM to inject knowledge into the LM.
- **K-BERT** [15] directly linked the related KG triplets with the sentence to inject the knowledge, which was fed to the LM together for the knowledge-enhanced representation.
- **KEPLER** [32] trained the LM as the knowledge embedding model, where the entity embedding was generated by the LM on the entity description.
- **K-Adapter** [31] designed a neural adapter for each kind of infused knowledge, and trained the adapters on different knowledge pretraining tasks.

*4.1.3 Training Details.* We implement KICP with Pytorch based on the pretrained BERT released by huggingface. [1] We use the "bert-base-chinese" version as *LM* for the Chinese dataset CN-QA and Math23K, and "bert-base-uncased" for the English dataset ComplexWebQuestions and FreebaseQA. For KA, the number of BERT layers of *Ad* is 12 (equal to *LM*), the dimension is 384 for hidden vector (half of *LM*) and 768 for output vector(equal to *LM*). The parameters of *Ad* are initialized with kaiming initialization. The implementations of KI and CR are available in section 3.3.

We pretrain the model on MLM task for 3 epochs with AdamW [18] as the KG is large enough. The batch size is set to 32, and the learning rate is 0.0005, which warms up over the first 10% steps, and then linearly decays. The masking probability for the MLM task is set to 0.15 in lesson 1 and lesson 3, and 0.3 in lesson 2 as the corpus in lesson 2 contains more repeated information. The masked tokens are processed following BERT.

In downstream QA tasks, for CN-QA, ComplexWebQuestions and FreebaseQA, we concatenate the question and each candidate answer as input to LM and implement the classifier *Pred* with MLP. CN-QA and ComplexWebQuestions are viewed as multi-label classification with more than one answers for each question and fine-tuned with binary cross entropy loss, and FreebaseQA is fine-tuned with cross entropy loss as single-label classification. For Math23K, we input the question into LM as encoder, and adopt GTS [36], an effective MWP solver, as decoder, which is fine-tuned with cross

entropy loss. The QA dataset is much smaller than the KG, thus we fine-tune for 30 epochs on CN-QA, ComplexWebQuestions and FreebaseQA, and 80 epochs on more difficult Math23K.

We run all experiments on a Linux server with two 2.20 GHz Intel Xeon E5-2650 CPUs and a Tesla K80 GPU.

## 4.2 Experimental Results

*4.2.1 Overall Results.* In this section, we compare KICP with all baselines. We use the F1 score (F1) and exact match score (EM) as metrics for multi-label datasets CN-QA and ComplexWebQuestions, and accuracy (ACC) for single-label dataset FreebaseQA. Math23K is evaluated with answer accuracy (ACC), i.e., the predicted expression is viewed correct if the computed answer equals the truth.

The results on four datasets are reported in Table 1. [2] We statistically test the improvement of KICP over baselines with paired t-test, and find the improvement to be significant with $p < 0.05$ (marked *). We can get the following observations from the results. First, KICP outperforms all baselines, which clearly demonstrates its effectiveness on knowledge learning and exploitation for QA tasks. Second, KICP performs better than K-Adapter, which has similar model but different pretraining task and dataset setting. The result shows that task and dataset have a significant influence on the pretraining outcome. Third, the knowledge-enhanced methods outperform the original BERT and RoBERTa in most cases, proving that knowledge is a key element in QA reasoning especially for KBQA. Last, knowledge injection does not bring much improvement and even negative effect on Math23K dataset. The reason may be that Math23K requires NLU more than knowledge, which may be affected by knowledge injection and thus hurts reasoning.

*4.2.2 Ablation Study.* Besides the widely studied entity relation knowledge stored in IDs, KICP further incorporates the attribute knowledge in diverse forms. Moreover, KICP designs the knowledge adapter module to reduce the negative impacts of the generated corpus. Therefore, in this section, we conduct ablation experiments to study the effectiveness of the two components (the curriculum reasoning will be investigated in detail in section 4.3). We introduce two variants of KICP: KICP-KA removes the knowledge adaptation module and directly trains the parameters of original LM, and KICP-ATT discards the attribute knowledge in KG and pretrains only on

---

[1]https://huggingface.co/transformers

[2]We do not evaluate KEPLER and K-Adapter on Math23K, as pretraining the two methods requires entity descriptions, which are unavailable on HowNet.

**Table 2: Performances on Easy and Hard Questions**

| Dataset | CN-QA | | FreebaseQA | |
| Difficulty | Easy | Hard | Easy | Hard |
|---|---|---|---|---|
| BERT | 0.633 | 0.603 | 0.920 | 0.891 |
| KICP | 0.676 | 0.634 | 0.933 | 0.907 |

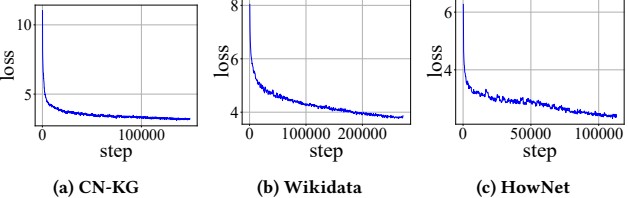

| (a) CN-KG | (b) Wikidata | (c) HowNet |

**Figure 3: Pretraining loss trend on three KGs in lesson 1.**

the entity relation knowledge. The results of the two variants are also reported in Table 1. [3] We can summarize the following conclusions from the results. First, the two variants both perform worse than KICP, which shows that the KA module could help address the shortcomings mentioned before, and the attribute knowledge is also quite useful in the KBQA tasks. Next, in CN-QA, KICP-ATT performs worse than KICP-KA, which means that exploitation of the attribute knowledge contributes more than the knowledge adaptation on this task. The result is reasonable since a large part of CN-QA requires attribute knowledge (about 45%). Last, KICP-KA performs worse than BERT in Math23K, which may be due to the reason that KICP-KA hurts the NLU ability of original LM in knowledge pretraining.

*4.2.3 Performance over Difficulty.* We also investigate the performance of KICP on questions with different difficulties to study the complex reasoning ability of the framework. We split CN-QA and FreebaseQA into easy questions (answerable with one knowledge triplet) and hard ones (requiring reasoning over multiple triplets). [4] We report the performances of KICP and BERT in Table 2 (F1 on CN-QA and ACC on FreebaseQA for simplicity). We have the following observations. First, it is a reasonable result that all methods perform much better on the easy questions than the hard ones. Second, KICP outperforms BERT on both easy and hard questions, showing that both easy and complex QA reasoning benefits from knowledge injection and exploitation. Next, the improvement on hard questions are larger in FreebaseQA. The reason may be that KICP are pretrained on corpus requiring more reasoning ability, which contributes to the higher performance in hard questions. However, in CN-QA the easy questions benefit more, which may result from the much larger proportion of easy questions benefiting from knowledge, and leads to a higher improvement.

### 4.3 Curriculum Reasoning Analysis

In this section, we investigate the feasibility and effectiveness of curriculum reasoning in KICP.

---

[3]The results of KICP-ATT on ComplexWebQuestions, FreebaseQA and Math23K are unavailable, as Wikidata and HowNet do not contain attribute knowledge.
[4]ComplexWebQuestions only contains hard questions and Math23K is a generative dataset which exploits knowledge implicitly and hard to distinguish the knowledge requirement and difficulty, so we do not conduct the experiment on the two datasets.

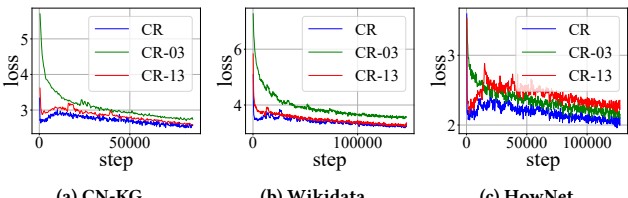

| (a) CN-KG | (b) Wikidata | (c) HowNet |

**Figure 4: Pretraining loss trend on three KGs in lesson 3.**

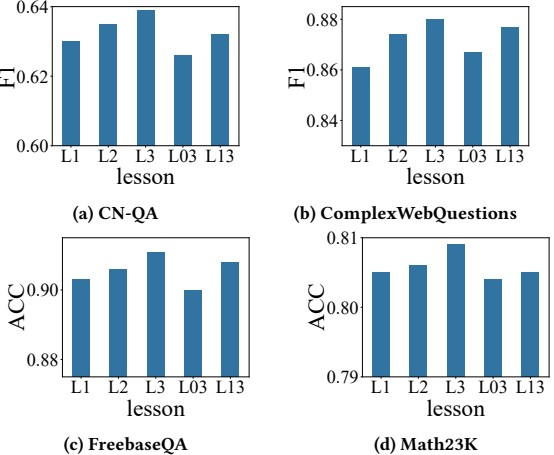

| (a) CN-QA | (b) ComplexWebQuestions |

| (c) FreebaseQA | (d) Math23K |

**Figure 5: Performances of LM pretrained for each lesson.**

*4.3.1 Loss of Curriculum Pretraining.* It is obvious that the corpus generated by the CR module is greatly different from the natural ones. To verify the feasibility of pretraining the LM with such corpus, we plot the trend of loss in pretraining. Due to the limited space, we report the lesson 1 results on three KGs in Figure 3. [5] From the figure, the loss keeps dropping and then gradually converges, which demonstrates that the generated corpus contains enough information to train the LM for knowledge learning, although it may seem odd compared with natural ones.

The CR module aims to reduce the difficulty of pretraining LM for complex reasoning in lesson 3. To investigate the effectiveness, we plot the loss trend in lesson 3 in Figure 4 with two variants: CR-03 trains on lesson 3 without previous two lessons, and CR-13 skips lesson 2. There are several observations from the figure. First, the loss of CR drops faster and finally reaches lower, proving that the curriculum setting could reduce the training difficulty by reaching a better initial state. Second, the trend of CR-03 is similar to lesson 1 in Figure 3, which may mean that the model first need to learn basic knowledge as in lesson 1 and then reasoning in CR-03. Third, the loss of CR and CR-13 has a short increase in the beginning which may be due to the higher difficulty of lesson 3 and the different data distribution between previous easier lesson. Last, CR-13 works better than CR-03 in CN-KG and Wikidata, showing that the LM can perform reasoning better with knowledge memorized. The exception in HowNet may be due to that HowNet mainly contains semantic information, which has been partially covered in LM.

*4.3.2 Performance of Curriculum Reasoning.* We also evaluate the effectiveness of CR on downstream QA tasks. Ideally, the LM would

---

[5]ComplexWebQuestions and FreebaseQA both use Wikidata as KG, so three KGs are included in total in experiments.

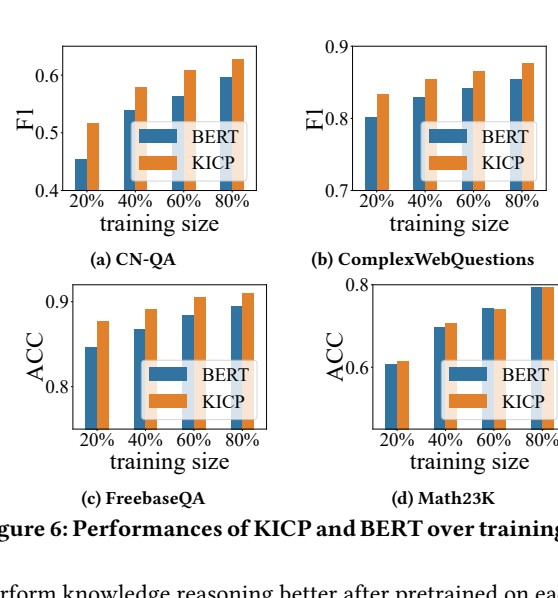

**Figure 6: Performances of KICP and BERT over training size.**

perform knowledge reasoning better after pretrained on each lesson. Therefore, we evaluate the LM finishing lesson 1, 2, 3 ("L1", "L2", "L3") on QA datasets in Figure 5 with CR-03 and CR-13 ("L03" and "L13") for comparison. We can get the following observations. First, performances of models keep increasing after finishing each lesson, which proves the above assumption. Next, the performance of L3 is much better than L03 and L13 (all pretrained on lesson 3), showing that the curriculum setting not only helps in convergence, but also promote the final outcome. Last, the performances of models on Math23K do not differ greatly. The reason may be that Math23K requires NLU more than knowledge, thus the knowledge pretraining has less effect on the QA performances.

### 4.4 Training Size Analysis

The pretrained LM aims to reduce the requirement of labeled data and improve the generalization, so the LM pretrained on the KG is expected to have a better performance than the original ones with limited labeled data. Therefore, we split the QA datasets with different training proportion (i.e., 20%, 40%, 60%, 80%) to evaluate performances of KICP and BERT. The results are demonstrated in Figure 6. From the figure, there are several observations. First, the performances of both KICP and BERT reasonably increase with more training samples. Next, although KICP outperforms BERT in all training settings, generally the differences are larger with less training data. The reason may be that the pretrained KICP could utilize the knowledge learned from KG and exploit less labeled data to learn the mapping from question to answer and achieve a good performance, while BERT needs to learn knowledge from the labeled data, which may be harder without enough data and result in worse performance.

### 4.5 Case Study

We demonstrate three typical cases by KICP and BERT on KBQA datasets in Table 3, and provide more in Appendix D. In case 1, BERT does not understand the knowledge about the lyricist of the song, and fails in the question, while KICP learns related knowledge in pretraining and correctly answer the question. In case 2, KICP

**Table 3: Cases of KICP and BERT**

**Case 1:** Who composed the song *Alexander's Ragtime Band* in 1911 ?
**KICP:** Irving Berlin (**correct**)
**BERT:** Woody Guthrie (**wrong**)

**Case 2:** Thomas Harris's 1988 novel *The Silence of the Lambs* was actually a sequel - what was the name of the first book in the series ?
**KICP:** *Red Dragon* (**correct**)
**BERT:** *Dubliners* (**wrong**)

**Case 3:** Which producer is responsible for *Pearl Harbour*, *Pirates of the Caribbean*, and *Armageddon* ?
**KICP:** Robert Mulligan (**wrong**)
**BERT:** John Ridley (**wrong**)
**Answer:** Jerry Bruckheimer

is capable of conducting multi-hop reasoning to find the complex relation between "Thomas Harris", "*The Silence of the Lambs*" and "*Red Dragon*" for the answer when the direct relation is unavailable, while BERT does not support such complex reasoning. In case 3, although both methods fail in the question, KICP predicts a closer answer which is also a producer with related knowledge, but BERT fails and makes an unrelated prediction.

## 5 CONCLUSION

In this paper, we proposed a general **K**nowledge-**I**njected **C**urriculum **P**retraining framework (KICP) to fully learn and exploit the KG for question answering, which could work with different detailed techniques for flexible application. We developed a general knowledge injection module to convert the KG into the pretraining corpus for LM with three key steps, and proposed a knowledge adaptation module to reduce the negative impacts of the gap between the generated and natural corpus by keeping the NLU ability of LM in knowledge learning. Furthermore, we designed a curriculum reasoning module to effectively pretrain the LM for human-like complex knowledge reasoning. Experimental results on four QA datasets demonstrated that the proposed KICP could achieve a more comprehensive learning and exploitation of KG for questions answering, and the curriculum setting could effectively reduce the pretraining difficulty and promote the outcome.

However, the proposed framework still had some limitations. First, the diversity of corpus generated by KICP was limited, and it would benefit pretraining if the generated corpus could be more similar to natural ones. Second, in the paper we mainly focused on the LM for language understanding, and we will generalize our framework to generative LM in the future. Last, KICP only exploited the KG as knowledge source, while there were much more types of knowledge to be studied.

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

## A  ARCHITECTURE OF KNOWLEDGE ADAPTER

The architecture of the KA module implemented in section 3.3 is demonstrated in Figure 7, where $s_i$ represents the semantic vector from the $i$th layer of the original language model $LM$, and $h_i$ represents the hidden knowledge-enhanced vector output of the $i$th layer of the knowledge adapter $Ad$. $AL$ means the adapter layer, $Trm^L$ and $Trm^K$ mean transformer blocks with different hidden dimension, and $Ln^I$ and $Ln^O$ mean linear models.

## B  DATASET STATISTICS

The statistics of the datasets in our experiments are available in Table 4. Note that ComplexWebQuestions and FreebaseQA use the same KG in pretraining (Wikidata). From the statistics, there are some interesting observations. First, only CN-KG contains both relation and attribute knowledge, and attribute knowledge counts for a large proportion in the KG (i.e., about 36%). Both Wikidata and HowNet only contain entity relation knowledge. Second, we split the questions in the QA datasets into simple ones (answerable with only one knowledge triplet) and hard ones (requiring complex reasoning over multiple triplets). Note that Math23K is a generative dataset, which exploits the knowledge implicitly, thus it is hard to distinguish the knowledge requirement and split the dataset. The CN-QA and FreebaseQA dataset both contain simple and hard questions, while ComplexWebQuestions only contains hard questions. The proportion of hard questions in CN-QA is much smaller than FreebaseQA, but CN-QA contains questions with more than one answers and requires both multi-hop and multi-object reasoning which have higher difficulty, while FreebaseQA has exactly one answer for each question and requires multi-hop reasoning only. Last, although the pretraining corpora are generated dynamically by KICP framework, we make a rough estimation of the corpus size based on the sampling and generation strategies. We can see that the sizes of generated corpora in lesson 2 and lesson 3 are the same as they are both based on the complex compositions following certain reasoning patterns, and the size of hard corpus (lesson 2 and 3) is larger than easy corpus (lesson 1) for Wikidata and HowNet as we consider two reasoning patterns (multi-hop reasoning and multi-object reasoning) and generate corpus for both of them. However, the size of hard corpus is not doubled in three KGs and even smaller than the easy one in CN-KG. The reason is that the one sentence in the hard corpus is composed of several knowledge triplets matching specific restrictions (e.g., multi-hop reasoning requires the intermediate entity to act as both the head and tail in two triplets) which reduces the number of acceptable triplets, while the simple corpus can be constructed using all triplets and thus has a larger size.

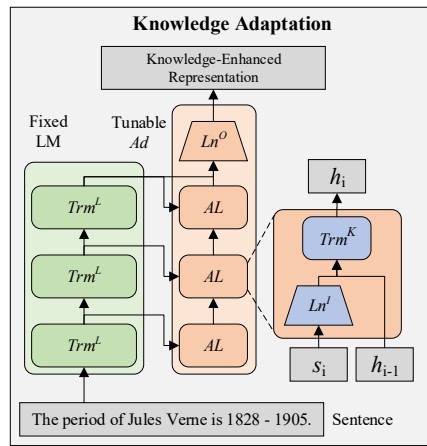

**Figure 7: An implementation of the knowledge adaptation module (KA) using BERT with additional inputs.**

## C  SAMPLES OF CORPUS

We demonstrate some samples of the constructed corpora for the three lessons of the CR module in Table 5. We place the unmasked version of each sentence on first line and masked one on second, and recover the split words for readability. The sentences are all in lower cases due to tokenization. We also provide related knowledge in the last two lines for lesson 3 for readability as some key information may be discarded. From these samples, we can get the following observations. First, the constructed sentences are similar to natural corpus to some extent with some unimportant differences. For example, "wilhelm friedrich kuhne member of royal society" in case 2 is similar to its natural form "wilhelm friedrich kuhne is a member of royal society" with differences on "is" and "a", which are unimportant for the meaning of the sentence. Second, all the head entities, tail entities and the relations may be masked as a whole to force knowledge learning (e.g., sample 1 to 3) according to the masking strategy. Third, following BERT, the masked token may be replaced with "[MASK]" or a random token ("orthogonal" in sample 4), or keep unchanged ("flag" in sample 5). Fourth, the complex corpus in lesson 2 and 3 may be constructed with two (sample 6) or three (sample 7) triplets, and considers both the multi-hop reasoning (sample 6, 7 and 9) and multi-object (sample 8 and 10) as mentioned in section 3.3, which increases the diversity of dataset and reasoning difficulties. Fifth, in lesson 2, sampled name of the same entity may be different for different triplets in the same sentence ("p:nsw" and "au-ns" in sample 6), but they are masked together, although the processing may be different ("divides into" in sample 6). Last, lesson 2 and lesson 3 share similar complex corpus construction methods (i.e., lesson 3 takes the final composition part of lesson 2), but lesson 2 gives all related knowledge triplets, which reduces the difficulty and serves as a preliminary of lesson 3.

## D  MORE CASES

We also provide more cases predicted by KICP and BERT on the KBQA datasets in Table 6 in addition to section 4.5. We classify these cases into three categories, i.e., the easy questions, hard questions, and wrong questions that both KICP and BERT fail. We can summarize the following observations. First, the easy questions

**Table 4: Statistics of Datasets**

| Dataset
KG | CN-QA
CN-KG | ComplexWebQuestions
Wikidata | FreebaseQA
Wikidata | Math23K
HowNet |
|---|---|---|---|---|
| #Questions | 13,041 | 13,544 | 15,811 | 23,162 |
| #Simple questions | 12,265 | 0 | 13,070 | / |
| #Hard questions | 776 | 13,544 | 2,741 | / |
| Avg. answer per question | 1.67 | 1.43 | 1 | 1 |
| #Entity | 1,477,923 | 397,133 | 397,133 | 237,861 |
| #Relations & attributes | 1,112 | 733 | 733 | 6 |
| #All triplets | 6,352,980 | 2,900,156 | 2,900,156 | 1,206,695 |
| #Relation triplets | 4,081,756 | 2,900,156 | 2,900,156 | 1,206,695 |
| #Attribute triplets | 2,271,224 | 0 | 0 | 0 |
| # Corpus for lesson 1 | 6,352,980 | 2,900,156 | 2,900,156 | 1,206,695 |
| # Corpus for lesson 2 | 1,806,861 | 3,128,153 | 3,128,153 | 1,356,960 |
| # Corpus for lesson 3 | 1,806,861 | 3,128,153 | 3,128,153 | 1,356,960 |

can be answered with only one knowledge triplets, which investigates whether the LM can memorize and exploit the knowledge. From the cases, KICP performs better than BERT. Next, the hard questions require reasoning over multiple knowledge facts. There are two typical mistakes in these cases, i.e., wrong answers (case 6 and 7) and failed prediction (case 5), which shows that the method may be not so capable of effective reasoning. Last, there are also questions mistakenly answered by KICP (case 8 and 9). In these cases, both the two methods make similar wrong prediction, which shows that there are still much room to improve for KICP, such as more reasoning patterns and more efficient knowledge learning and exploitation.

**Table 5: Samples of the Constructed Corpus in the CR Module**

| Lesson | | Samples |
|---|---|---|
| Lesson 1 | (1) | [CLS] sir frederick ashton nationality united kindom [SEP]
[CLS] [MASK] [MASK] [MASK] nationality united kindom [SEP] |
| | (2) | [CLS] wilhelm friedrich kuhne member of royal society [SEP]
[CLS] wilhelm friedrich kuhne member of [MASK] [MASK] [SEP] |
| | (3) | [CLS] republic of maldives used money maldivian rufiyah [SEP]
[CLS] republic of maldives [MASK] [MASK] maldivian rufiyah [SEP] |
| | (4) | [CLS] sarbogard district time euro time [SEP]
[CLS] sarbogard district time [MASK] orthogonal [SEP] |
| | (5) | [CLS] first hellenic republic flag flag of greece [SEP]
[CLS] [MASK] [MASK] [MASK] flag flag of greece [SEP] |
| Lesson 2 | (6) | [CLS] collaroy plateau based in p : nsw [SEP] au - ns divides into gundagai shire council [SEP] collaroy plateau based in divides into gundagai shire council [SEP]

[CLS] collaroy plateau based in p : nsw [SEP] au - ns [MASK] into gundagai shire council [SEP] collaroy plateau based in [MASK] [MASK] gundagai shire council [SEP] |
| | (7) | [CLS] star fox 64 3d part of the series star fox ( virtual boy ) [SEP] starfox ( virtual boy ) characters fox makuraudo [SEP] fox mccloud recording by ohara takashi [SEP] star fox 64 3d part of the series characters recording by ohara takashi [SEP]

[CLS] star fox 64 3d part of the series [MASK] fox [MASK] [MASK] [MASK] ) [SEP] starfox ( virtual boy ) [MASK] fox makuraudo [SEP] [MASK] [MASK] [MASK] [MASK] recording by ohara takashi [SEP] star fox 64 3d part of the series [MASK] recording by ohara takashi [SEP] |
| | (8) | [CLS] spannarhyttan timezone utc + 2 : 00 [SEP] spannarhyttan timezone utc + 1 : 00 [SEP] spannarhyttan timezone utc + 2 : 00 utc + 1 : 00 [SEP]

[CLS] spannarhyttan timezone utc [MASK] [MASK] : [MASK] [SEP] spannarhyttan [MASK] [MASK] utc + 1 : 00 [SEP] spannarhyttan ##unes ##zone [MASK] [MASK] 133 : [MASK] utc + 1 : 00 [SEP] |
| Lesson 3 | (9) | [CLS] theobald ziegler working at on lake the rhine [SEP]
[CLS] theobald ziegler working at on lake [MASK] [MASK] [SEP]
( [CLS] theobald ziegler working at strassbourg [SEP]
[CLS] strassbourg on lake the rhine [SEP] ) |
| | (10) | [CLS] ferrieres , somme shares border with ailly - sur - somme pont - de - metz [SEP]
[CLS] ferrieres , somme [MASK] [MASK] [MASK] ailly - sur - somme pont - de - metz [SEP]
( [CLS] ferrieres , somme shares border with ailly - sur - somme [SEP]
[CLS] ferrieres , somme shares border with pont - de - metz [SEP] ) |

**Table 6: More Cases Predicted by KICP and BERT**

| Category | Cases |
|---|---|
| Easy | **Case 1:** Aberystwyth lies on which bay ? 
 **KICP:** Cardigan (**correct**) 
 **BERT:** Blaenau Gwent (**wrong**) |
| | **Case 2:** In *Alice in Wonderland*, who wanted to decapitate anyone who offended her ? 
 **KICP:** Queen of Hearts (**correct**) 
 **BERT:** Daisy Fay (**wrong**) |
| | **Case 3:** Who wrote the thriller novel *Birds of Prey* ? 
 **KICP:** Wilbur Smith (**correct**) 
 **BERT:** Ludwig von Mises (**wrong**) |
| | **Case 4:** Io, Europa, Ganymede and Callisto are all moons of which planet in our solar system ? 
 **KICP:** Jupiter (**correct**) 
 **BERT:** Pluto (**wrong**) |
| Hard | **Case 5:** What kind of money does the country with the nation anthem *Du gamla, Du fria* use ? 
 **KICP:** Swedish Krona (**correct**) 
 **BERT:** / (**wrong**) |
| | **Case 6:** What form of government is used in the country that uses Chilean Peso ? 
 **KICP:** Presidential system \| Unitary state (**correct**) 
 **BERT:** Presidential system \| Unitary state \| Patrimonial monarchy (**wrong**) |
| | **Case 7:** What is the nationality of the author of *The Little Prince* ? 
 **KICP:** France (**correct**) 
 **BERT:** America (**wrong**) |
| Wrong | **Case 8:** Which comedy actor played Charlie Bind in the 1964 film *Carry on Spying* ? 
 **KICP:** Peter Hinwood (**wrong**) 
 **BERT:** Peter Hinwood (**wrong**) 
 **Answer:** Charles Hawtrey |
| | **Case 9:** What team did Drogba play for that won the 2014 Coupe de France Final championship ? 
 **KICP:** Piast Gliwice (**wrong**) 
 **BERT:** Germinal Beerschot (**wrong**) 
 **Answer:** En Avant de Guingamp |

