# OpenReview forum: "A Knowledge-Injected Curriculum Pretraining Framework for Question Answering"
_ACM.org/TheWebConf/2024/Conference — TheWebConf24_

### Official Review · Reviewer_TuU5 · 2023-10-30

**Novelty:** 4
**Technical Quality:** 5

**Review:**

This paper proposes KICP, a knowledge base question-answering approach focusing on the pretraining stage. Experiments on four datasets demonstrate that KICP outperforms several baselines.

- There are several works focusing specifically on KG-based pretraining for knowledge-intensive tasks, some pointers would include [1-4]. While it is not strictly necessary (but recommended) to compare with them, it is at least helpful to discuss the differences of KICP and these approaches.

- Which base language model did baselines (ERNIE, K-BERT, etc.) use? Is it the same with KICP? Please clarify this in the paper to support the claim for a fair comparison.

- It would be helpful to run baselines and KICP over multiple random seeds for the experiments in Table 1 since some of the performance gaps may seem marginal. In addition, it would be nice to include statistical significance test results to claim for "significant" improvements.

- I suggest expanding the ablation study. Overall KICP involves a lot of moving parts and design choices, but the ablation study only looked at 2 of them.

- On a side note, in the current atmosphere of large language models, I'm not sure if knowledge base augmentation is still the go-to paradigm for knowledge-intensive tasks. Maybe no need to include them as baselines, but it would be helpful to discuss and acknowledge in the related work state-of-the-art approaches such as retrieval augmentation, multi-LM collaboration, search engine integration, and more, as valid approaches to compare and contrast with the proposed approach.

[1] Lu, Yinquan, et al. "KELM: Knowledge Enhanced Pre-Trained Language Representations with Message Passing on Hierarchical Relational Graphs." ICLR 2022 Workshop on Deep Learning on Graphs for Natural Language Processing. 2022.

[2] Feng, Shangbin, et al. "Factkb: Generalizable factuality evaluation using language models enhanced with factual knowledge." Proceedings of the 2023 Conference on Empirical Methods in Natural Language Processing, Singapore. Association for Computational Linguistics. 2023.

[3] Peters, Matthew E., et al. "Knowledge Enhanced Contextual Word Representations." Conference on Empirical Methods in Natural Language Processing and the 9th International Joint Conference on Natural Language Processing (EMNLP-IJCNLP). 2019.

[4] Agarwal, Oshin, et al. "Knowledge Graph Based Synthetic Corpus Generation for Knowledge-Enhanced Language Model Pre-training." Proceedings of the 2021 Conference of the North American Chapter of the Association for Computational Linguistics: Human Language Technologies. 2021.

**Questions:**

please see above

**Reviewer Confidence:**

3: The reviewer is confident but not certain that the evaluation is correct

**Scope:**

3: The work is somewhat relevant to the Web and to the track, and is of narrow interest to a sub-community

---

### Official Review · Reviewer_2duh · 2023-11-24

**Novelty:** 5
**Technical Quality:** 5

**Review:**

Pros:
- This paper revolves around knowledge graphs, directly utilizing the knowledge graph as a dataset for training language models. This approach aims to enable the model to adapt to various knowledge graph-based applications. To validate the effectiveness of the method, this paper takes knowledge graph question-answering as an application example. It implements knowledge graph question-answering based on pre-trained language models and compares and analyzes it with traditional pre-trained language models like BERT, RoBERTa as well as knowledge-enhanced language models such as ERNIE, etc.
- This paper proposes to utilize Curriculum Learning to help the model learn more effectively by gradually increasing the difficulty or complexity of training samples for obtaining the ability of complex knowledge reasoning.
- This paper proposes to add the AD module for adapting the model to minimize the semantic gap between the generated sentences from KG and the natural language.

Cons:
- There is a lack of experimental results compared with KGQA based on knowledge graph representation learning techniques. For example, EmbedKGQA is another technical line, but its method (based on ComplEx to learn embeddings of entities) leverages the structure of the knowledge graph and learns semantic relationships within the graph. Transforming the knowledge graph into serialized text sequences for language model training raises the question of whether this approach has advantages on KG.

**Questions:**

Can the authors provide some explanations for the comments above?

**Reviewer Confidence:**

3: The reviewer is confident but not certain that the evaluation is correct

**Scope:**

4: The work is relevant to the Web and to the track, and is of broad interest to the community

---

### Official Review · Reviewer_7xA8 · 2023-11-28

**Novelty:** 4
**Technical Quality:** 5

**Review:**

This paper presents an approach called KICP to improve KG learning for KBQA tasks, which consists of three steps: knowledge injection (KI), knowledge adaptation (KA) and curriculum reasoning. The approach addresses some of the limitations of existing knowledge-enhanced LMs by adding some novel ideas based on knowledge adaptation and curriculum learning. The quality of writing is high and the method is described in detail with formalisations. The claims are validated through an experiment using four datasets, and the results are compared with state-of-the-art baselines.


Pros:
- The proposed approach addresses some of the shortcomings of the previous knowledge-enhanced LMs using techniques such as the usage of tunable knowledge adapters while keeping the original LM weights intact and curriculum learning for multi-hop reasoning with three different levels of difficulties.
- The proposed approach is evaluated on four KBQA datasets, and it outperforms the relevant baselines, including both standard LMs and knowledge-enhanced LMs such as KEPLER and K-Adapter.

Cons:
- The proposed work is mainly focused on addressing multi-hop reasoning. It is not clear how generalisable the approach is to other types of reasoning required for complex KBQA.

**Questions:**

- It seems pretraining corpora for curriculum reasoning is mainly focused on 3 lessons with the complexity of multi-hop reasoning with composition learning. However, KBQA, in general, can have other complex reasoning aspects such as temporal reasoning, geographical reasoning, superlative/comparative reasoning, aggregations, taxonomic reasoning, etc. How generalisable is your approach for these different reasoning types? How easy is to generate corpora covering different types of reasoning with curriculum reasoning?
- Regarding the point that LMs have strong NLU capabilities but don’t work well on knowledge applications, another alternative that has been widely used to solve the issue is to use Retrieval Augmented Generation (RAG). What would be the benefits of using a pretraining approach compared to a retrieval approach?

**Reviewer Confidence:**

3: The reviewer is confident but not certain that the evaluation is correct

**Scope:**

3: The work is somewhat relevant to the Web and to the track, and is of narrow interest to a sub-community

---

### Official Review · Reviewer_NKzy · 2023-11-29

**Novelty:** 4
**Technical Quality:** 5

**Review:**

This paper presents a method for knowledge-based question answering that leverages language models and knowledge graphs. The method is based on different steps, notably a knowledge injection one in which the LM is fed with KG triples; a domain adaptation one and the inclusion of inference based on compositional rules.

The positive aspect of this work is mostly the extensive use of KG to improve the performance of LMs (BERT-like) on QA tasks.

However, there are some negative aspects; first of all, the work "forgets" the progress that has been done in QA and LMs since the BERT-like models introduction. As it can be easily checked with a generative model, the question that seems to have motivated the work "What is the period of the author of Off on a Comet ?" is pretty easily solved in GPT-like models, as all the questions in Table 3. Thus, some claims in the paper feel unsubstantiated: "How to enable the LMs with such complex reasoning is a challenging problem"; "Pretraining the LM on the corpus may hurt NLU ability and work badly on natural texts" require at least some citations or examples.
The paper is difficult to follow because of language; some sentences are difficult to understand as they are too long and convoluted. All over the paper the authors use "triplets" instead of "triples": a triplet in English is used to indicate 3 identical babies from the same mother.

**Questions:**

"First, existing methods rewriting the KG into corpus often depend on specific techniques and resources for effective generation, while our
method is a general framework": as far as I can understand, you also have to make choices at KA and CR steps? In which sense the framework is generalizable?

"existing methods focus more on improving language understanding with related knowledge but seldom consider the human-like complex reasoning ability of LMs" can you link to some examples and the related methods?

**Ethics Review Description:**

no need

**Reviewer Confidence:**

3: The reviewer is confident but not certain that the evaluation is correct

**Scope:**

3: The work is somewhat relevant to the Web and to the track, and is of narrow interest to a sub-community

---

### Decision · Program_Chairs · 2024-01-22

**Decision:**

Accept

**Comment:**

This paper proposes a knowledge-based QA method integrating language models and knowledge graphs, leveraging KG triples injection, domain adaptation, and compositional rule-based inference.
 Advantages include extensive use of KGs to enhance language model performance, the proposed knowledge adaptation and curriculum module, along with experiments. KICP outperforms baselines, addressing previous knowledge-enhanced language model shortcomings with tunable knowledge adapters and curriculum learning for multi-hop reasoning.
 The authors addressed some of concerns from the reviewers in the rebuttal, such as the generalisation
  to other complex KBQA reasoning types beyond just multi-hop reasoning, the transformation of the KG into serialized text sequences ,the lacking experimental results compared to KGQA with KG representation learning techniques, and they also conducted additional experiments with EmbedKGQA, and additional ablation experiments
 The authors' response and additional experiments demonstrate a commitment to enhancing the manuscript based on reviewers' feedback. Please include all clarifications and additional experiments in the camera-ready version.